# RulePrompt: Weakly Supervised Text Classification with Prompting PLMs and Self-Iterative Logical Rules

## ABSTRACT

Weakly supervised text classification (WSTC), also called zero-shot or dataless text classification, has attracted increasing attention due to its applicability in classifying a mass of texts within the dynamic and open Internet environment, since it requires only a limited set of seed words (label names) for each category instead of labeled data. With the help of recently popular prompting pre-trained language models (PLMs), many studies leveraged manually crafted and/or automatically identified verbalizers to estimate the likelihood of categories, but they failed to differentiate the effects of these category-indicative words, let alone capture their correlations and realize adaptive adjustments according to the unlabeled corpus. In this paper, in order to let the PLM better understand each category, we at first propose a novel form of rule-based knowledge using logical expressions to characterize the meanings of categories. Then, we develop a prompting PLM-based approach named RulePrompt for the WSTC task, consisting of a rule mining module and a rule-enhanced pseudo label generation module, plus a self-supervised fine-tuning module to make the PLM align with this task. Within this framework, the inaccurate pseudo labels assigned to texts and the imprecise logical rules associated with categories mutually enhance each other in an alternative manner, establishing a self-iterative closed loop of knowledge (rule) acquisition and utilization, with seed words serving as the starting point. Extensive experiments validate the effectiveness and robustness of our approach, which outperforms state-of-the-art weakly supervised methods. Importantly, our approach yields interpretable category rules, proving its advantageous for disambiguating easily-confused categories.

## CCS CONCEPTS

• **Information systems** → **Web mining**; **Clustering and classification**; • **Computing methodologies** → **Learning paradigms**.

## KEYWORDS

weak supervision, text classification, seed word, pre-trained language model, prompt, logical rule, rule mining, pseudo label

**ACM Reference Format:**
Anonymous Author(s). 2018. RulePrompt: Weakly Supervised Text Classification with Prompting PLMs and Self-Iterative Logical Rules. In *Proceedings of Make sure to enter the correct conference title from your rights confirmation email (Conference acronym 'XX)*. ACM, New York, NY, USA, 9 pages. https://doi.org/XXXXXXX.XXXXXXX

## 1 INTRODUCTION

With the rapid development of Internet, an abundance of textual content is produced across news media and social networks. It is significant and challenging to classify these texts into predefined categories, especially when up-to-date labeled data are hard to access due to the dynamic and open nature of Internet. Consequently, there has been a growing interest in weakly supervised text classification (WSTC) [15, 21, 28, 29, 36, 37], also known as zero-shot or dataless text classification [3, 4, 12, 13, 20–22, 26, 27, 30, 31, 33, 38], which only requires a limited set of seed words (label names) for each category.

Recently, the proliferation of prompting pre-trained language models (PLMs) greatly bolstered the WSTC task, but their performances still lag behind supervised methods [29]. Since no labeled data are available as evidence, relying solely on seed words for grasping category meanings proves inadequate. In previous research, many approaches either provided manual verbalizers of categories or automatically discover them based on word embedding similarity. Taking them as additional knowledge, some studies estimated category likelihoods by tapping into the generative capability of PLMs [37, 38], and others leveraged PLM's effective vector representations to calculate the similarity or entailment between texts and categories [24, 27]. However, most of them failed to differentiate the effects of these category-indicative words (abbreviated as indicative words). Although NPPrompt [38] did calculate and utilize the weights of them, their roles in classification remained independent of each other and lacked adaptive adjustments based on the current corpus, so cannot accommodate ever-changing Internet environment.

However actually, the effect of each category-indicative word varies. Certain words can determine the category on its own, like the label names, while others need to be used cooperatively to distinguish between easily-confused categories. For example, the word "*penalty*" itself cannot signify the "*Sports*" category, but when combined with "*goal*", the text is likely to talk about a football match. Conversely, an additional word "*company*" could imply the "*Society*" category rather than "*Sports*". Therefore, a simplistic set of indicative words is not enough to cover the full meanings of categories. Instead, logical operations such as conjunction and disjunction are appropriate to capture the correlation of these words as enriched knowledge for weakly supervised classification. Fortunately, the flexibility of prompting PLMs just offers an opportunity to apply these logical rules in the template to achieve precise semantic representation of categories.

It is obvious that logical rules are difficult to set manually as prior knowledge, but they can be mined from preliminarily categorized

texts with the aid of pseudo labels generated by the PLM. Furthermore, the mined rules and pseudo labels can mutually enhance each other in an alternative way, establishing a self-iterative closed loop for knowledge acquisition and utilization, with seed words as the starting point. That poses two main challenges: (1) When inaccurate pseudo labels are available, how to identify candidate category-indicative words using the PLM and build correlations among them by means of logical rules to characterize each category? (2) With imprecise logical rules, how to effectively transform them into the PLM template for classification by handling each logical operator individually, and then update the pseudo label assigned to each text?

To address these issues, this paper at first proposes a novel kind of rule-based knowledge in the form of logical expressions for category understanding in WSTC. Each category is represented by a disjunctive normal form, where indicative words serve as atomic propositions. Specifically, a single disjunctive term (one-literal clause) denotes strong and self-explanatory indicative words, while a clause of conjunctive form depicts the synergistic effect of weak and polysemous indicative words.

Based on this, a prompting PLM-based approach for text classification is developed, through iteratively updating both the pseudo label of each text and the logical rule of each category. That is realized mainly via two modules, rule mining and rule-enhanced pseudo label generation. The former first extracts signal words from each text by the PLM, and regards these words as a transaction of the relevant category decided by the current pseudo labels. For each category, we mine frequent 1-itemsets (items) and 2-itemsets respectively from specific subsets of transactions, and construct the disjunctive normal form. In the latter, the current logical rule for each category is injected into three PLM-based models, each providing a different perspective. Then, a new pseudo label is generated for each text via integrating the results of these models. In addition, in each iteration, the PLM can be fine-tuned with a self-supervised loss to better align with the task requirements.

In summary, the contributions of this paper include:

- To the best of our knowledge, this is the first attempt to differentiate the effects of category-indicative words in the WSTC task and characterize category meanings through logical rules, thereby establishing a new paradigm for knowledge representation in this field.
- A novel approach leveraging prompting PLMs is presented to make the pseudo labels of texts and the logical rules of categories enhance each other iteratively. That facilitates a sufficient fusion of automatically generated rule-based knowledge and unlabeled data.
- Comprehensive experiments conducted on multiple real datasets demonstrate the effectiveness and interpretability of our approach. It consistently outperforms state-of-the-art weakly supervised methods, and yields intuitive logical rules for categories to avoid confusion.

## 2 RELATED WORK

### 2.1 Weakly Supervised Text Classification

Weakly supervised text classification (WSTC) demands minimal seed information, such as label names or extended keywords for each category, thereby significantly reducing the cost of text annotation. At an early stage, some researchers used auxiliary knowledge bases like Wikipedia to establish the semantic correlation between texts and labels [3, 26]. Subsequently, topic-model based methods emerged [4, 12, 13, 30, 31], which inferred category-aware topics from a limited set of seed words. In the last few years, neural methods has gained prominence [20, 21, 28, 33, 36]. They trained neural classifiers using pseudo labels of texts, often relying on generated pseudo-texts or PLMs to detect category-indicative keywords. For example, LOTClass [22] used label names as the only seed words, and introduced BERT for category understanding.

In recent time, prompting-based methods [6, 10, 23] have been extensively developed for the WSTC task. A lot of work harnessed the strong generative capability of PLMs with instruction template for classification. For instance, NPPrompt [38] used initial word embeddings by PLM to automatically construct verbalizers without manual design or unlabeled corpus, and estimated probability distribution over categories through weighted sum of these words. PromptClass [37] introduced a noise-robust method to iteratively self-train text classifiers and update pseudo labels, employing two fine-tuning strategies of PLMs to improve the quality of pseudo labels. WDDC [32] utilized generated words for the [MASK] token as supervision signals, and proposed a latent variable model to train a word distribution learner and a text classifier simultaneously. Other approaches explored the effective vector representation power of prompting PLMs. PESCO [27] incorporated label descriptions into predefined prompts, formulating the WSTC task as a neural matching problem. Meanwhile, LIME [24] used large textual entailment models trained with external data to suggest seed words and infer text labels.

Although these methods have demonstrated inspiring performance, a gap still exists when compared to fully supervised methods. Due to the absence of labeled data, there is a need to automatically extract and apply additional knowledge from unlabeled data during the classification process. Existing methods just relied on a set of category-indicative words, but have not taken the varying effect of these words into account, which leads to imprecise category understanding.

### 2.2 Logical Rules for Natural Language Processing Tasks

Recently, there has been increasing research interest in the integration of logical rules into natural language processing tasks, aiming to improve the interpretability of neural network models.

Hu et al. [11] proposed a teacher-student framework combining deep neural networks with first-order logic rules, and transformed the structured information of logic rules into the weights of neural networks. TALLOR [14] addressed the named entity tagging problem by using a small set of seed logical rules as weak supervision, and further selected new accurate logical rules based on a hand-tuned threshold. PTR [9] incorporates logic rules to encode human prior knowledge and composes several manually designed sub-prompts into final task-specific prompts. PRBoost [34] viewed top-$k$ predictions from the [MASK] token as candidate rules through the disjunction operation. They generated these rules from

large-error instances based on a few labeled data, and then used human-selected rules to generate weak labels for model training.

However, most of these previous work required seed rules as initial supervision or human feedback when selecting accurate rules. In contrast, our approach focuses on the WSTC task, and establishes self-iterative closed loop for the acquisition and utilization of logical rules, eliminating the need for human intervention. Additionally, while existing PLM-based methods primarily employed one operator when composing decision rules, we consider both the disjunction and conjunction operators to distinguish the strength and effect of indicative words, enabling a more precise understanding of categories.

## 3 PRELIMINARIES

In this section, we formulate the task of weakly supervised text classification (WSTC), and briefly introduce prompting PLMs as well as two roles of them as the foundation of our approach.

### 3.1 Problem Formulation

Given a corpus of unlabeled texts $D = \{D_1, \ldots, D_N\}$ and a set of target categories $Z = \{z_1, \ldots, z_K\}$ with a label name $l(z)$ for each $z \in Z$, weakly supervised text classification (WSTC) aims to assign a category label $z(d)$ to each text $d$. Following the extremely weak supervision setting [28], only the sole label surface name of each class is used as supervision here, without other seed words.

### 3.2 Prompting PLMs for Estimating Likelihoods

Prompt-based tuning applies cloze-style tasks to tune PLMs. A prompt is composed of a template $\mathcal{T}(\cdot)$ and a set of selected words $\mathcal{V}$. We can fill each text $d$ into the template $\mathcal{T}(\cdot)$ to obtain the prompt input $\mathcal{T}(d)$. For example, for the text classification task on news, the prompt can be written as

$$\mathcal{T}(d) = d \text{ It is about [MASK] news.} \tag{1}$$

In vanilla prompt engineering, the verbalizer, i.e., an injective mapping function $\phi : Z \rightarrow \mathcal{V}$, links the category set and the set of selected words. Then, at the masked position, we can calculate the likelihood for each category via word probability distributions.

$$P(z|d) = P([MASK] = \phi(z) \mid \mathcal{T}(d)). \tag{2}$$

Recently, A lot of work studied for a verbalizer with richer label words to represent the category. Typically, NPPropmt [38] constructs a $K$-nearest-neighbor verbalizer, through searching over the whole vocabulary $\mathcal{V}$ for the top-$k$ nearest words to the label name of $z$ in the embedding space of the PLM, denoted as $\mathcal{M}(z)$.

$$\mathcal{M}(z) = \underset{v \in \mathcal{V}}{\text{Top}-K_0}\{\text{sim}(\text{emb}(v), \text{emb}(l(z)))\}, \tag{3}$$

where $\text{emb}(v)$ and $\text{emb}(l(z))$ are the embeddings of word $v$ and label name $l(z)$ respectively, and $\text{sim}(\cdot)$ means cosine similarity.

Then, we get the unnormalized probability for each category:

$$Q(z|d) = \sum_{v \in \mathcal{M}(z)} w(v, l(z)) \cdot \Theta([MASK] = v \mid \mathcal{T}(d)), \tag{4}$$

where $\Theta$ is the kernel smoothing on logits instead of probability, and $w(v, l(z))$ is the weight of the word $v$ on the label name $l(z)$,

defined in the softmax form:

$$w(v, l(z)) = \frac{\exp(\text{sim}(\text{emb}(v), \text{emb}(l(z))\})}{\sum_{v' \in \mathcal{M}(z)} \exp(\text{sim}(\text{emb}(v'), \text{emb}(l(z))))} \tag{5}$$

Besides, NPPrompt uses more than one keywords for some categories. The final score is calculated as follows:

$$Q(z|d) = \max_{v \in \Phi(z)} Q(v|d) \tag{6}$$

where $\Phi(z)$ contains all keywords for category $z$, and $Q(v|d)$ is computed similar to Equation 4, replacing the category $z$ by one of its indicative words $v$ and the label name $l(z)$ just by itself $v$.

### 3.3 Prompting PLMs for Getting Signal Words

In addition to estimating category likelihoods, some work utilized prompting PLMs to generate words which can summarize the content of the given text. That is also based on the probability distribution over $\mathcal{V}$, and can be used to get better supervision information than the words themselves appearing in the text. Formally, given a threshold $K_1$, for each text $d$, top $K_1$ words with higher logits can be seen as the signal words of $d$, denoted as $SW(d)$.

$$SW(d) = \underset{v \in \mathcal{V}}{\text{Top}-K_1}\{P([MASK] = v \mid \mathcal{T}(d))\}, \tag{7}$$

## 4 METHOD

In this section, we at first define logical rules of categories as a new kind of knowledge. Based on this, the framework of RulePrompt is presented followed by details of the three key modules.

### 4.1 Logical Rules of Categories

In this paper, we propose a novel kind of rule-based knowledge representation for categories as additional weak supervision information in text classification. It takes automatically mined category-indicative words as atomic propositions, and build their correlation through logical expressions with disjunction and conjunction operators. Specifically, each category can be represented by a disjunctive normal form.

*Definition 4.1 (Logical Rules of Categories).* The meaning of each category $z$ can be represented by a logical rule as follows:

$$r(z) = \Big(a_1 \vee \cdots \vee a_S\Big) \vee \Big((b_{11} \wedge b_{12}) \vee \cdots \vee (b_{T1} \wedge b_{T2})\Big) \tag{8}$$

where both $a_j(1 \leq j \leq S)$ and $b_{j1}, b_{j2}(1 \leq j \leq T)$ are indicative words of the category $z$. The rule can be divided into two sub-rules. The first $S$ words are strong and can indicate the category on its own, so they are connected directly by the disjunction operator and compose the disjunctive sub-rule, denoted as $r^d(z)$. The last $2T$ words are comparatively weak and need to act together to imply the category, so they are firstly paired with the conjunctive operator, and then combined by disjunction. That is called the conjunctive sub-rule and denoted as $r^c(z)$. It is reasonable to think the two parts of the rule characterize a category in different views.

### 4.2 Framework

On this basis, we propose a novel prompting PLM-based approach for the WSTC task as shown in Figure 1. At first, as the starting point with only label names, we leverage a classical zero-shot prompting method using PLM [38] to generate the initial pseudo labels

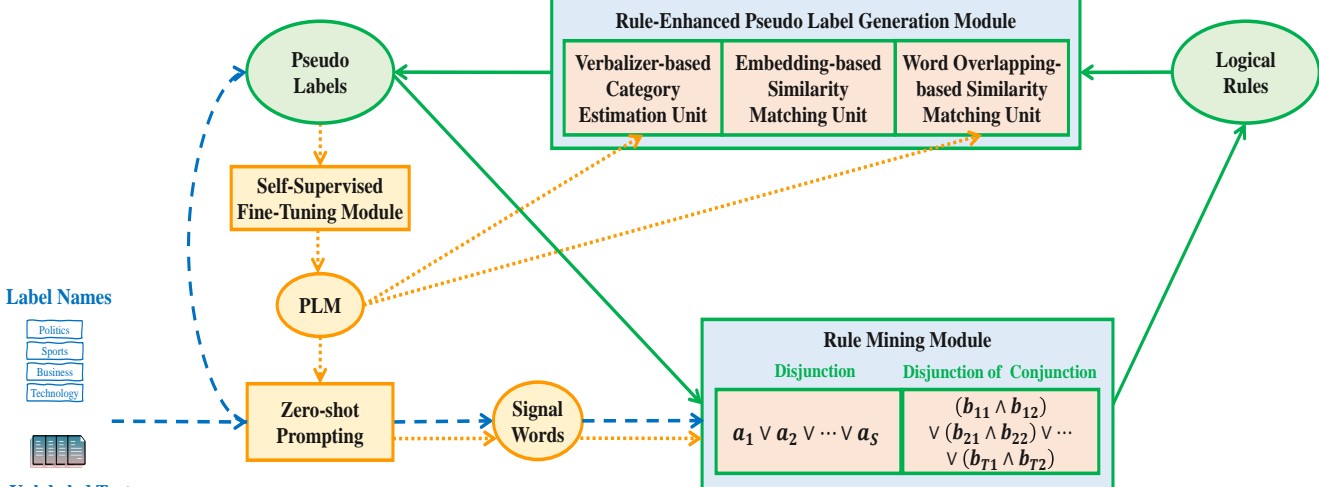

Figure 1: Framework of the Proposed Approach RulePrompt.

and the signal words of texts (blue dash line). Then, the approach enters the self-iteration between pseudo labels and category knowledge (logical rules) through mutual enhancement (green solid line). Meanwhile, the PLM is gradually optimized by self-supervised fine-tuning to better support the main iteration above (yellow dotted line). To this end, three modules are designed.

In the rule mining module, based on the current pseudo labels with confidence scores, we cluster the unlabeled texts assigned to each category into three sets. Then, with the signal words of each text obtained by PLM, frequent 1-itemsets (items) and 2-itemsets of each category are mined from the first two confident sets respectively, which composes the disjunctive normal form of the logical rule for each category.

In the rule-enhanced pseudo label generation module, we incorporate the current logical rules into three prompting PLM-based classification models from different perspectives to update pseudo labels. On the one hand, the words in the disjunctive sub-rule with higher support is directly used to obtain richer verbalizers in the generation-based model. On the other hand, the whole rule is injected into templates to derive texts for similarity-based classification. That is realized in two views, global embedding similarity and local word overlapping. Finally, these results are averaged to get new pseudo labels of texts.

In addition, in order to make the PLM accommodate this specific task, the self-supervised fine-tuning module is executed after each time pseudo labels are generated, employing self-supervised loss over high-confidence texts.

## 4.3 Rule Mining Module

In weakly supervised setting, only label names are not adequate to reflect the meanings of categories. Thanks to the strong generative and representation capability of prompting PLMs, it is feasible to utilize the pseudo labels and signal words of texts to furthermore understand categories and enrich the prior knowledge. Since pseudo labels are imperfect, to mitigate error propagation, the selection of texts and signal words should be restricted to those with high

confidence. Inspired by [2], we define the confidence score (of the pseudo label) of a text as

$$conf(d) = P(z_{(1)}|d) - P(z_{(2)}|d) \qquad (9)$$

where $z_{(1)}$ and $z_{(2)}$ respectively denote the first and the second most probable label for text $d$ computed by the prompting PLM. Compared to the highest probability, the difference value gives a better indication of how confident the PLM regards the current unique prediction.

However, for each category $z$, the numbers of texts appropriate to extract strong words and weak words are hard to determine, so we adaptively cluster the texts assigned to $z$ into three sets by K-means, based on the confidence scores. These texts with excellent, good and poor quality, are denoted as $D_z^1$, $D_z^2$ and $D_z^3$ respectively.

For the signal words of texts, the set $SW(d)$ computed by Equation 7 needs to be further filtered to guarantee their quality. We utilize the whole corpus to pursue the speciality of signal words for the text, which we think can better imply the assigned category as well. The new unnormalized probability can be calculated as:

$$P'([\text{MASK}] = v \mid \mathcal{T}(d)) = \frac{P([\text{MASK}] = v \mid \mathcal{T}(d))}{\frac{1}{N} \sum_{d' \in D} P([\text{MASK}] = v \mid \mathcal{T}(d'))}, \quad (10)$$

Then, we select the top $K_2$ signal words with higher logits as the strong signal words, denoted as $SSW(d)$.

$$SSW(d) = \underset{v \in \mathcal{V}}{\text{Top}-K_2}\{P'([\text{MASK}] = v \mid \mathcal{T}(d))\}, \qquad (11)$$

Next, we use frequent pattern mining [1, 8, 25] to obtain representative rules of categories. For $D_1^i$ and $D_2^i$, we treat each text as a transaction and each strong signal word of it as an item of the transaction. Then, we at first pay attention to the most confident set $D_z^1$ to mine frequent 1-itemsets (items) with a pre-defined support threshold $h_1$, which compose the disjunctive sub-rule of $z$, as each of them alone is enough to indicate a category. The support of a

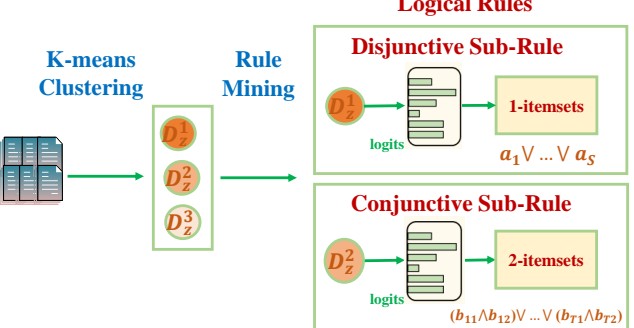

**Figure 2: Rule Mining Module.**

word $a$ in $D_z^1$ is calculated as:

$$sup(a, D_z^1) = \frac{\sum_{d \in D_z^1} I_1(a, d)}{|D_z^1|} \qquad (12)$$

where $I_1(a, d)$ is an indicator function expressing whether $a$ is in the transaction of $d$,

$$I_1(a, d) = \begin{cases} 1, a \in SSW(d), \\ 0, a \notin SSW(d). \end{cases} \qquad (13)$$

Moreover, for the set $D_z^2$ with moderate confidence scores, we mine 2-itemsets given another threshold $h_2$ to construct the conjunctive sub-rule. Although these words cannot represent a category individually, their co-occurrence in the set of strong signal words should also be captured. The support of a 2-itemset $b = b_1 \land b_2$ is calculated as:

$$sup(b, D_z^2) = \frac{\sum_{d \in D_z^2} I_2(b, d)}{|D_z^2|} \qquad (14)$$

where $I_2(b, d)$ is another indicator function expressing whether both $b_1$ and $b_2$ are in the transaction of $d$,

$$I_2(b, d) = \begin{cases} 1, b_1 \in SSW(d) \land b_2 \in SSW(d), \\ 0, b_1 \notin SSW(d) \lor b_2 \notin SSW(d). \end{cases} \qquad (15)$$

Besides, we need to exclude those pairs containing words also appearing in the frequent 1-itemsets for any other category in $D_z^2$, which would bring confusion.

## 4.4 Rule-Enhanced Pseudo Label Generation Module

In this subsection, we present the reverse direction of the iteration, i.e., how to inject the mined logical rules of categories into the pseudo labels generation process. Considering the diverse capabilities of PLMs and the distinct roles that logical rules play within them, three units from three perspectives are designed to compute the probability of each text belonging to each category. The final results are obtained by averaging the outputs from the three units.

*4.4.1 Verbalizer-based Category Estimation Unit.* Since label names are too limited to characterize categories, the indicative words in our logical rule can be naturally used to expand the verbalizers in classical zero-shot prompting model (Equation 2). In view of the strict requirement of verbalizers, for each category, we only use the words

in the first half of the disjunctive sub-rule according to their support values. The expanded set is written as $\Phi'(z) = \{l(z), a_1, a_2, \ldots, a_{\frac{S}{2}}\}$ similar to the manually crafted set of keywords in Equation 6. Besides, inspired by NPPrompt [38], the top-$K_0$ closest words to each of them are also used to complement the verbalizers. For a verbalizer word $v \in \Phi'(z)$, we can get the probability $Q(v|d)$ and then take the maximum value among all verbalizers as aggregated probability $Q(z|d)$ similar to Equation 6, as all of these words can imply the category independently.

Noticing that $Q$ is an unnormalized probability, we use the softmax function to transform the value between 0 and 1, to get the probability $P_1(z|d)$ from the first perspective.

$$P_1(z|d) = \frac{\exp(Q(z|d))}{\sum_{z' \in Z} \exp(Q(z'|d))} \qquad (16)$$

*4.4.2 Embedding-based Similarity Matching Unit.* To conduct a similarity-based matching between a text and a category through prompting PLM, an intuitive idea is to put the logical rule of each category into the template as in Equation 1. However, the expression of conjunction and disjunction in the [MASK] token is not like natural language, which would affect the semantic understanding of the PLM. Hence, we handle each indicative word separately instead and combine them in different ways for disjunction and conjunction.

For the disjunctive sub-rule, we directly calculate embedding-based similarity between a text $d$ and a category $z$ as weighted sum of the similarity between $d$ and each word $a$ in the sub-rule of $z$:

$$ES^d(d, z) = \frac{\sum_{a \in r^d(z)} sup(a, D_z^1) \cdot sim(f(d), g(a))}{S}, \qquad (17)$$

where $sup(a, D_z^1)$ is the support of the word $a$ in $D_z^1$, $f(d)$ is the sentence embedding of text $d$, and $g(a) = f(\mathcal{T}(a))$ is the embedding of the template after removing "d" and replacing [MASK] with $a$.

While for the conjunctive sub-rule, besides the outer disjunction operation can be handled in the same way, the similarity between $d$ and each 2-itemset $b = b_1 \land b_2$ is computed through the weighted composition of vectors instead of similarity scores.

$$ES^c(d, z) = \frac{\sum_{b \in r^c(z)} sup(b, D_z^2) \cdot sim(f(d), g'(b))}{T}, \qquad (18)$$

where $sup(b, D_z^2)$ is the support value of the 2-itemset $b$ in $D_z^2$, and the embedding variant $g'(b)$ can be computed as the weighted sum of the embedding vectors of the either conjunctive term of $b$.

$$\begin{aligned} g'(b) = \quad & \frac{sup(b_1, D_z^2)}{sup(b_1, D_z^2) + sup(b_2, D_z^2)} \cdot f(\mathcal{T}(b_1)) \\ & + \frac{sup(b_2, D_z^2)}{sup(b_1, D_z^2) + sup(b_2, D_z^2)} \cdot f(\mathcal{T}(b_2)) \qquad (19) \end{aligned}$$

At last, the embedding-based similarity $ES(d, z)$ between $d$ and $z$ is defined as the maximum value for the two sub-rules, and regarded as the probability $P_2(z|d)$ from the second perspective.

$$P_2(z|d) = ES(d, z) = \max(ES^d(d, z), ES^c(d, z)) \qquad (20)$$

*4.4.3 Word Overlapping-based Similarity Matching Unit.* Following the idea of PRBoost [34], besides embedding-based similarity matching in a global view, we also consider the word overlapping-based similarity in a local view, leveraging PLMs' capability of generating

signal words from texts once again. In this way, the rule is no longer inserted into the [MASK] token, but instead occupies the position of the input text in the template as an independent sentence. Consequently, indicative words within a rule can be connected by the word "and" to form a coherent sentence, regardless of the actual logical operator. However, the use of the word "or" as a connector is not consistent with typical human speech patterns.

We still deal with two sub-rules separately. For the disjunctive sub-rule, the overlapping of strong signal words is computed as:

$$OS^{\mathrm{d}}(d, z) = \frac{SSW(d) \cap SSW(\mathcal{T}(And(r^{\mathrm{d}}(z))))}{K_2}, \tag{21}$$

where $And(\cdot)$ is a transformation function from a logical rule to a sentence, which connects the indicative words of the rule with "and". For instance, $And(r^c(z)) = $ "$a_1$ and $a_2$ and ... and $a_S$".

For the conjunctive sub-rule, as the involved indicative words are weaker, the matching process should be more strict. Hence, we divide the sub-rule into two parts alternately, construct the sentence separately, and take the maximum of the similarity scores:

$$OS^c(d, z) = \max\left(\frac{SSW(d) \cap SSW(\mathcal{T}(And(r^{c1}(z))))}{K_2},\right.$$
$$\left.\frac{SSW(d) \cap SSW(\mathcal{T}(And(r^{c2}(z))))}{K_2}\right), \tag{22}$$

where $r^{c1} = \{b_{11}, b_{12}, b_{31}, b_{32}, \ldots\}$ and $r^{c2} = \{b_{21}, b_{22}, b_{41}, b_{42}, \ldots\}$.

Finally, the similarity of word overlapping between a text $d$ and a category $z$ is defined as the sum over both sub-rules. Then, the corresponding probability from the third perspective is obtained through the softmax function.

$$OS(d, z) = OS^{\mathrm{d}}(d, z) + OS^c(d, z), \tag{23}$$

$$P_3(z|d) = \frac{\exp(OS(d, z))}{\sum_{z' \in Z} \exp(OS(d, z'))}. \tag{24}$$

To get a final predictive probability, the three scores from different perspectives are averaged together to supplement each other.

$$P(z|d) = (P_1(d, z) + P_2(d, z) + P_3(d, z))/3 \tag{25}$$

Based on this, the pseudo label of a text can be assigned to the category with the maximum probability.

## 4.5 Self-Supervised Fine-Tuning Module

Although prompting PLMs are strong enough to assist deriving classification results in various manners, they are not specially designed for the WSTC task. Therefore, we introduce self-supervised fine-tuning into the closed loop, which uses the PLM's current prediction $P_1(d, z)$ to refine PLM itself, gradually enabling it to adapt to the specific task. Specifically, we adopt self-supervised entropy [18] as the loss function to sharpen the probability distribution of category assignments generated by the PLM. That can maximize the potential of the PLM and mitigate the accumulation and propagation of errors during the model training process. Given the inaccuracy of pseudo labels, we just select a majority of texts (denoted as $D'$) with tolerable predictive probability for fine-tuning. Formally, the loss is defined as follows:

$$L = \sum_{d \in D' \subset D} \sum_{z \in Z} -P_1(z|d) \log P_1(z|d). \tag{26}$$

**Table 1: Dataset Statistics.**

| Dataset | # Texts | # Classes | Classification Type | Imbalance |
|---------|---------|-----------|---------------------|-----------|
| AGNews | 120000 | 4 | News Topics | 1.0 |
| NYT | 31997 | 9 | News Topics | 27.09 |
| IMDB | 25000 | 2 | Review Sentiment | 1.0 |

The fine-tuning is conducted after each main iteration updates the pseudo labels of texts, and when rule mining module is executed in the next iteration, new signal words derived by the fine-tuned PLM can be used. The overall algorithm is shown in Algorithm 1.

---

**Algorithm 1** RulePrompt

---

**Input:** An unlabeled text corpus $D$; a set of categories $Z$ with label names; a pre-trained language model (PLM) $M$.

**Output:** The category label $z(d)$ of each text $d \in D$.

1: Obtain initial pseudo labels $z^{(0)}(d)$ via probability distribution $P(z|d)$ for each text $d \in D$ through NPPrompt with Equation 4;
2: **for** $i = 1$ to $Iter$ **do**
3:     Obtain the confidence score of each text with Equation 9;
4:     Obtain strong signal words $SSW(d)$ for each text $d \in D$ through the PLM $M$ with Equation 11;
5:     **for all** category $z \in Z$ **do**      ▷ Rule Mining
6:         cluster the texts assigned to $z$ into $D_z^1, D_z^2, D_z^3$ based on their confidence scores;
7:         Mine 1-itemsets from $D_z^1$ with Equation 12;
8:         Mine 2-itemsets from $D_z^2$ with Equation 14;
9:         Compose logical rule $r^{(i)}(z)$ according to Definition 4.1;
10:     **end for**
11:     **for all** text $d \in D$ **do**      ▷ Pseudo Label Generation
12:         Obtain new pseudo label $z^{(i)}(d)$ via probability distribution $P(z|d)$ with Equation 25;
13:     **end for**
14:     Fine-tune the PLM $M$ with Equation 26;      ▷ Fine-Tuning
15: **end for**
16: **return** $z^{(Iter)}(d)$;

---

## 5 EXPERIMENTS

In this section, we first introduce datasets, baselines and experimental settings in the experiments. Then, overall results are presented to demonstrate the effectiveness and robustness of the proposed approach. Finally, we analyze the importance of components by ablation study, as well as the key hyperparameter, iteration number.

The experiments were performed on NVIDIA A40 GPUs, and implemented based on an open-source toolkit OpenPrompt [5].

### 5.1 Experimental Setup

*5.1.1 Datasets.* We use three popular datasets from the Internet for evaluation. The statistics of them are shown in Table 1.

- AGNews [35] is a news article dataset from AG's corpus.
- NYT [35] contains news articles written and published by New York Times, covering abundant news topics.
- IMDB [19] is for sentiment classification of movie reviews.

**Table 2: Label Names and Templates for RulePrompt.**

| Dataset | Label Names | Template |
|---------|-------------|----------|
| AGNews | politics, sports, business, technology | A [MASK] news: $d$ |
| NYT | business, politics, sports, health, education, estate, arts, science, technology | Topic: [MASK] $d$ |
| IMDB | good, bad | $d$ In summary, the film was [MASK] . |

*5.1.2 Baselines.* We compare our approach with the following weakly supervised methods. The first two are seed-driven methods, which require at least three keywords for each category as input, and others belong to emerging PLM-based methods.

- **WeSTClass** [21] generates pseudo labels based on word embeddings and obtains the final classifier via self-training.
- **ConWea** [20] generates pseudo labels based on the contextualized representations of keywords, and trains a text classifier to further expand the keyword sets.
- **LOTClass** [22] utilizes the pre-trained BERT to find indicative keywords, which are directly used for category understanding and feature representation learning.
- **XClass** [28] expands indicative words for category-oriented representations, and generates pseudo labels for fine-tuning a text classifier via clustering.
- **ClassKG** [33] builds a keyword graph with co-occurrence relation, and generates pseudo labels via a self-trained subgraph annotator, used to update keywords iteratively.
- **NPPrompt** [38] constructs verbalizers based on initial word embeddings by PLM, and estimates probability distribution over categories via weighted sum of these words.
- **PromptClass** [37] utilizes zero-shot prompting to generate pseudo labels and improves the quality of them through two fine-tuning strategies of PLMs.

Besides, we also inspect a fully supervised method, which uses BERT classifier with fine-tuning based on the labels in the training set. It can be regarded as an upper-bound for WSTC methods.

*5.1.3 Experimental Settings.* We use the same input label name of each category as previous work, and list them as well as the template for each dataset in Table 2. As prompt-based methods are relatively robust with PLMs [29], we follow previous work [27, 38] to choose RoBERTa-large [16] as our PLM.

In regard to getting signal words and strong signal words of texts, we set $K_1 = 100$ and $K_2 = 20$. In the process of frequent pattern mining, we set support thresholds $h_1 = h_2 = 0.1$ for AGNews and IMDB, 0.05 for imbalanced NYT. As only top 1-itemsets and 2-itemsets can enter the rule, these thresholds are insensitive and can thus be a low value. The maximum numbers of terms in the disjunctive sub-rule and conjunctions in the conjunctive sub-rule are both $S = T = 10$. In the embedding-based similarity matching unit, we choose Roberta-SimCSE [7] as the sentence encoder.

In the self-supervised fine-tuning module, we train 5 epochs in each iteration for NYT and IMDB, for AGNews that is larger, the number of epochs is 8. We use AdamW [17] as the optimizer. The number of full iterations *Iter* is unified to 3 across all datasets. Besides, the proportion of texts used for fine-tuning is set to 80%.

Following previous work, we also use Micro-F1 and Macro-F1 as the evaluation metrics. The results of baselines are quoted from [34]

with missing values marked as "-". Since NPPrompt uses more than one keyword on some datasets in its original setting, we re-run it with their codes[1] using only the label names for fair comparison.

## 5.2 Overall Results

The overall results of RulePrompt, its variant without fine-tuning, and baseline methods are shown in Table 3.

It is evident that our model consistently outperforms baselines for all datasets, and almost catch up with the supervised methods on IMDB. That certifies the role of logical rules of categories in assisting prompting PLMs to understand the topics of texts, compared with independent category-indicative words. In addition, the advantage over PromptClass highlights the importance of the mutual enhancement of pseudo labels and logical rules, as they are both imperfect at the starting point. Although RulePrompt exhibits a slight gap with ConWea on the Macro-F1 metric in the imbalanced NYT dataset, which is caused by the amplification of categories with small samples, our approach is more stable across all datasets.

As to the largest AGNews dataset, RulePrompt surpasses the variant without fine-tuning, but in case of NYT and IMDB, adopting a fixed PLM is more appropriate. That can be explained by the dataset size. When there is sufficient evidence available for each category, even if unlabeled, it becomes feasible to refine the PLM to accommodate the specific task and dataset, with the help of self-iterative logical knowledge of categories. Conversely, when the dataset is small, its particularities are harder to summarize and the PLM is better to retain its initial state trained by large-scale Internet data.

For an interpretability analysis, in the AGnews dataset, we observe that for the category "politics", the words "war" and "palestinian" are mined as 1-itemsets to form the disjunctive sub-rule, while "world ∧ foreign" and "diplomatic ∧ geopolitical" are identified as 2-itemsets, forming the conjunctive sub-rule. These rules align with common intuitions and significantly contribute to a more comprehensive representation of their respective categories.

Furthermore, in the NYT dataset, the word "architecture" is found within rules associated with two different categories: "Estate" and "Arts". It is paired with "residential" and "apartments" for the former, but "museum" and "cultural" for the latter. That exemplifies the ability of our approach to disambiguate easily-confused categories.

## 5.3 Ablation Study

The ablation results for the two main modules are shown in Table 4. In order to make the role of each component more prominent, the experiments were carried out in the first iteration (denoted as RulePrompt-1), thus without self-supervised fine-tuning.

**In terms of rule mining.** The variants include removing the conjunctive sub-rule (−Conj), and mining rules from all texts without

---

[1]https://github.com/XuandongZhao/NPPrompt

Table 3: Overall Results on Three Datasets Measured by Micro-F1 and Macro-F1. The Best Scores are Marked in Bold.

| Methods | AGnews | | NYT | | IMDB | |
|---|---|---|---|---|---|---|
| | Micro-F1 | Macro-F1 | Micro-F1 | Macro-F1 | Micro-F1 | Macro-F1 |
| WeSTClass | 0.823 | 0.821 | 0.683 | 0.570 | 0.774 | - |
| ConWea | 0.746 | 0.742 | 0.817 | **0.715** | - | - |
| LOTClass | 0.869 | 0.868 | 0.671 | 0.436 | 0.865 | - |
| XClass | 0.857 | 0.857 | 0.790 | 0.686 | - | - |
| ClassKG | 0.881 | 0.881 | - | - | 0.888 | 0.888 |
| NPPrompt | 0.692 | 0.628 | 0.720 | 0.596 | 0.939 | 0.939 |
| PromptClass (RoBERTa+RoBERTa) | 0.895 | 0.895 | - | - | 0.906 | 0.906 |
| PromptClass (ELECTRA+ELECTRA) | 0.884 | 0.884 | - | - | 0.931 | 0.931 |
| RulePrompt | **0.897** | **0.896** | 0.822 | 0.699 | 0.940 | 0.940 |
| RulePrompt without Fine-Tuning | 0.842 | 0.838 | **0.826** | 0.702 | **0.941** | **0.941** |
| Fully Supervised | 0.940 | 0.940 | 0.943 | 0.899 | 0.945 | - |

Table 4: Results of Ablation Study for One Iteration. The Best Scores are Marked in Bold.

| Methods | AGnews | | NYT | | IMDB | |
|---|---|---|---|---|---|---|
| | Micro-F1 | Macro-F1 | Micro-F1 | Macro-F1 | Micro-F1 | Macro-F1 |
| RulePrompt-1 ($-$Conj) | 0.853 | 0.850 | 0.821 | 0.694 | 0.938 | 0.938 |
| RulePrompt-1 ($-D_z$) | 0.502 | 0.423 | 0.748 | 0.653 | 0.933 | 0.933 |
| RulePrompt-1 ($-U1$) | 0.763 | 0.761 | 0.600 | 0.542 | 0.912 | 0.912 |
| RulePrompt-1 ($-U2$) | 0.853 | 0.850 | 0.818 | 0.688 | 0.937 | 0.937 |
| RulePrompt-1 ($-U3$) | 0.852 | 0.849 | 0.820 | 0.693 | 0.935 | 0.935 |
| RulePrompt-1 | **0.854** | **0.851** | **0.823** | **0.700** | **0.940** | **0.940** |

clustering-based set division ($-D_z$). At first, the lack of conjunction part will lower the performance. That confirms the discrepancy among indicative words on characterizing category meanings, and the combined effect of relatively weaker words cannot be neglected. Besides, when the rules are mined from the whole corpus, the accuracy is distinctly declined. That can be attributed to the inaccurate pseudo labels which decides the mining object. Therefore, the confidence scores for the predicted labels is vital to help choose appropriate texts to search for rules in an adaptive way.

**In terms of rule-enhanced pseudo label generation.** The variants contain the methods without either of the three units respectively. For most cases, the full approach performs the best. That reflects different capabilities of the PLM as well as the different manners of logical rules enhancing the PLM. Since it is hard to determine which is better beforehand, averaging the predictive results of them to supplement each other is a good choice.

### 5.4 Analysis of Iteration Number

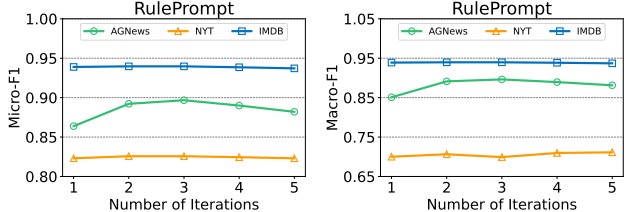

Figure 3: Results with Varied Number of Iterations.

We vary the number of full iterations *Iter* from 1 to 5 for all datasets. Figure 3 shows the values of two metrics for RulePrompt. It can be seen that the performance shows a trend of first rising and then declining, and reaches optimal after about three iterations. That is nearly consistent across all datasets, and thus indicates the robustness of this setting as well as our approach.

An exception appears for the imbalanced NYT when Macro-F1 is examined. That coincides with the observation in the overall results and proves the requirement of our approach for a certain number of unlabeled texts of each category to mine precise knowledge.

## 6 CONCLUSION

Addressing the limitations of relying solely on seed words (label names) for supervision in weakly supervised text classification task, this paper explores a kind of novel knowledge representation to characterize category meanings, which facilitates the effective integration of knowledge and unlabeled corpus. The proposed logical rules for categories can be automatically mined based on the pseudo labels of texts and iteratively self-optimized through mutual enhancement with them. Thanks to the enriched symbolic knowledge, the potential of prompting PLMs are further exploited in terms of generative capability and semantic representations, which is realized by incorporating the PLM into the rule-based iteration process. With this framework, RulePrompt exceeds the SOTA weakly supervised methods, and the logical rules we extract are intuitive and provide valuable guidance by disambiguating easily-confused categories. For future work, we will strengthen the expressiveness of the category rules, such as the adding negation operator. Additionally, more effective iteration strategies are also worth studying.

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
