# OpenReview forum: "RulePrompt: Weakly Supervised Text Classification with Prompting PLMs and Self-Iterative Logical Rules"
_ACM.org/TheWebConf/2024/Conference — TheWebConf24 Oral_

### Official Review · Reviewer_QeZV · 2023-11-17

**Novelty:** 4
**Technical Quality:** 4

**Review:**

The paper proposes a text classification approach for a weakly
supervised scenario. The approach combines frequent pattern mining (to
find representative words and word pairs for categories in texts) and
prompting pre-trained language models (to assign pseudo-labels and
extract signal words).

The approach is compared against seven competitors which it beats on
three datasets. However, PESCO [27] (cited in Line 200) is not among
them, though it also outperforms many of the competitors. For example,
on the AG News dataset, PESCO has an accuracy of 89.6 compared to 86.4
for LOTClass. Also, other approaches are compared on other/further
datasets, e.g., DBpedia, Yahoo Answers, or Amazon. For comparison it
would be crucial to use more of these common datasets and also compare
against PESCO. Otherwise, the benefits of the proposed method over the
state of the art are hard to judge.

The idea of combining LLMs with rule-based approaches is appealing and
I appreciate that the paper tries to pursue this goal. However, I find
the claim misleading that the approach uses logical rules. Basically,
frequent words and word pairs are used to compute text similarities
(using embeddings, cf. Eq. 17 and 18). Also, in Sec. 4.4.3 all words
are connected by "and" (cf. l 584). Thus, I could not find any
application of logics. Even if I regard this as "rule-based", the
paper would need to justify why a "rule-based knowledge representation
for categories" is novel. Disjunctions of (conjunctions of) words are
nothing special and have been used before (just think of Boolean
search). Finally, the ablation study should evaluate whether the word
pairs really bring some benefits – how would the approach perform, if
those words were treated like the individual words?

Suggestions for improvement:
- You might consider writing "Web" instead of "Internet", since that
  better fits to the observations you describe.
- The effect of the rules and the combination of the signal words is
  quite difficult to judge. Please provide some representative
  examples of rules which were generated by the proposed approach. The
  example in lines 790 to 793 is not convincing, since "palestinian"
  and "war" are quite restrictive words and I wonder why, for
  instance, the word "politician" is not among the words.
- The text contains many grammar issues and is at times difficult to
  understand. Please invest some time to improve the writing of the
  paper.

**Questions:**

- (Notwithstanding how the rules are actually used:) Why can in each
  rule only two words be joined by a conjunction and why are arbitrary
  conjunctions of words not allowed? This seems rather restrictive.
- How did you choose the support thresholds? How sensitive is the
  approach to other choices?

**Reviewer Confidence:**

2: The reviewer is willing to defend the evaluation, but it is likely that the reviewer did not understand parts of the paper

**Scope:**

2: The connection to the Web is incidental, e.g., use of Web data or API

---

### Official Review · Reviewer_6haf · 2023-11-22

**Novelty:** 5
**Technical Quality:** 6

**Review:**

The authors propose a rules-based and prompt-guided methodology, RulePrompt to tackle (extremely) weakly supervised text classification. The framework consists of rule-mining and rule-enhancing pseudo label generation modules. Pseudo-labels are refined via an iterative process, making the weakly supervised signal stronger. The results indicate RulePrompt is SOTA or on-par with three widely used datasets.

The framework is well-motivated and justified. While it could be somewhat complicated at first glance, the authors are able the clearly articulate each equation, idea, etc.

The results are somewhat convincing. The framework (completely) outperforms other baselines on 2/3 datasets (AGnews and IMDB), although it isn't by much (0.895 & 0.895 -> 0.897 & 0.897 for AGnews and 0.939 & 0.939 -> 0.941 & 0.941 for IMDB). The authors have a nice ablation study that not only looks at all of their components individually, but also across all datasets used. I believe experiments on more datasets are warranted, since RulePrompt is, at best, barely better than other frameworks. Nevertheless, it is definitely comparable with SOTA techniques. While not warrented, I believe the authors should've done a case study so that they could identify and potential patterns from incorrect predictions.

**Questions:**

NA

**Reviewer Confidence:**

3: The reviewer is confident but not certain that the evaluation is correct

**Scope:**

3: The work is somewhat relevant to the Web and to the track, and is of narrow interest to a sub-community

---

### Official Review · Reviewer_KiMf · 2023-11-24

**Novelty:** 4
**Technical Quality:** 5

**Review:**

The paper introduces RulePrompt, an innovative approach aiming to overcome the limitations of weakly supervised text classification reliant solely on seed words. It proposes a novel method to represent category meanings using automatically mined logical rules derived from pseudo labels of texts, iteratively self-optimized to enhance the understanding of categories. By integrating prompting Pre-trained Language Models (PLMs) into the rule-based iteration process, RulePrompt effectively harnesses symbolic knowledge, improving generative capability and semantic representations. Experimentally, RulePrompt consistently outperforms state-of-the-art weakly supervised methods, providing intuitive logical rules that aid in disambiguating confusing categories, especially on larger datasets. The conclusion highlights the need for future work to enrich rule expressiveness and develop more effective iteration strategies.

**Strengths**

- The paper introduces a novel method of deriving logical rules from pseudo labels, enhancing weakly supervised text classification beyond seed words.
- Comprehensive experiments demonstrate RulePrompt's consistent outperformance of existing methods, showcasing its effectiveness.
- The incorporation of PLMs into the iterative rule-based process leverages the potential of these models, improving semantic representations.

**Weaknesses**
- The method involves various steps, including iterative self-optimization and PLM integration, potentially raising computational complexity.
- While performing well, the approach's effectiveness might vary concerning specific datasets or domain-specific texts not covered in the evaluation.

**Questions:**

How does the proposed method's computational complexity scale with larger datasets or more complex rules?

**Reviewer Confidence:**

2: The reviewer is willing to defend the evaluation, but it is likely that the reviewer did not understand parts of the paper

**Scope:**

3: The work is somewhat relevant to the Web and to the track, and is of narrow interest to a sub-community

---

### Official Review · Reviewer_BcjZ · 2023-11-26

**Novelty:** 6
**Technical Quality:** 6

**Review:**

### Quality:

The paper presents a new approach to the Weakly supervised text classification (WSTC) task that combines logical rules and prompting pre-trained language models. The authors provide a thorough description of their methodology and experimental setup. The method design and evaluation seem sound to me.

### Clarity:

The paper is well-written and easy to follow.

### Originality:
The approach presented in this paper is relatively new. Although the concrete modules (e.g., prompting PLMs and enrichment of supervision) borrow insights from previous work in WSTC, the overall framework that incorporates logical rules to improve WSTC is novel. There is also some novelty in generating pseudo labels based on embeddings and rules.

### Significance:

The significance of this work lies in its potential to improve WSTC, which is an important task in text classification. The authors demonstrate that their approach outperforms existing methods on popular datasets, indicating that it could be a valuable tool.

### **Pros**:
- A relatively novel approach to WSTC that combines logical rules and prompting pre-trained language models
- Thorough explanation and sound designs of methodology and evaluation
- The proposed method demonstrates effectiveness and robustness on popular datasets over strong baselines

### **Cons**:
- The evaluation would benefit from incorporating large language model (LLMs) baseline results or at least discuss how LLMs could assist the WSTC task, though it is acceptable for a paper on WSTC to not compare to LLMs as prior work in WSTC usually assumes a small encoder model as the classifier. Some relevant studies can be found in the reference list below.
- The experiments should better use more datasets. I found the current three datasets used to be acceptable but they are indeed on the easier side of text classification tasks. Also, better to report the sensitivity of the method (e.g., via standard deviation over several runs) as the method seems to be a bit complex with iterative loops.
- (Minor suggestion) It looks like the referenced paper "PromptClass: Weakly-Supervised Text Classification with Prompting Enhanced Noise Robust Self-Training" has been updated with a different title "PIEClass: Weakly-Supervised Text Classification with Prompting and Noise-Robust Iterative Ensemble Training"

Reference:
* Meng et al. “Generating Training Data with Language Models: Towards Zero-Shot Language Understanding.” NeurIPS (2022).
* Ye et al. “ZeroGen: Efficient Zero-shot Learning via Dataset Generation.” EMNLP (2022).

**Questions:**

Please address the "cons" raised in my main review.

**Reviewer Confidence:**

4: The reviewer is certain that the evaluation is correct and very familiar with the relevant literature

**Scope:**

4: The work is relevant to the Web and to the track, and is of broad interest to the community

---

### Decision · Program_Chairs · 2024-01-22

**Decision:**

Accept (Oral)

**Comment:**

The authors discuss an approach that combined LLM and rule-based methods to overcome the limitations of weakly supervised text classifications. More specifically, the methods combines frequent pattern mining to find representative words and word pairs for categories in texts and prompting pre-trained language models to assign pseudo-labels and extract signal words. The method is thoroughly evaluated, and the empirical results demonstrated its performance.